# FIPER: Generalizable Factorized Fields for Joint Image Compression and Super-Resolution

## Abstract

In this work, we propose a unified representation for Super-Resolution (SR) and Image Compression, termed **Factorized Fields**, motivated by the shared principles between these two tasks. Both SISR and Image Compression require recovering and preserving fine image details—whether by enhancing resolution or reconstructing compressed data. Unlike previous methods that mainly focus network architecture, our proposed approach utilizes a basis-coefficient decomposition to explicitly capture multi-scale visual features and structural components in images, addressing the core challenges of both tasks. We first derive our SR model, which includes Coefficient Backbone and Basis Swin Transformer for generalizable Factorized Fields. Then, to further unify these two tasks, we leverage the strong information-recovery capabilities of the trained SR modules as priors in the compression pipeline, improving both compression efficiency and detail reconstruction. Additionally, we introduce a merged-basis compression branch that consolidates shared structures, further optimizing the compression process. Extensive experiments show that our unified representation delivers state-of-the-art performance, achieving an average relative improvement of 204.4% in PSNR over the baseline in Super-Resolution (SR) and 9.35% BD-rate reduction in Image Compression compared to the previous SOTA.

## 1 Introduction

Single Image Super-Resolution (SISR) aims to reconstruct high-quality images from low-resolution counterparts. Specifically, the key lies in accurately restoring fine details and reconstructing the correct arrangement of visual features. Thus, geometric correspondences or repetitive patterns, such as stripes, grids, or textures, are commonly used for evaluation due to their rich details that are crucial to image fidelity. Early CNN-based approaches Lim et al. (2017b); Dong et al. (2014)laid the foundation for SISR, which was later enhanced with GAN-based methods Ledig et al. (2017a) for improved perceptual realism. Follow-up Transformer-based networks Chen et al. (2021) address non-local dependencies, with subsequent Swin Transformer-based approaches sparking tremendous advancements Conde et al. (2022)Chen et al. (2023d)Chen et al. (2024), which then inspire more and more complex, delicate designs of heavy network architectures in SISR. However, these prior works have primarily focused on network architecture design rather than addressing the capability of representation. The visual patterns and the inherent nature of image content structure that play an influential role in SISR have not been explicitly considered in the representation learning process. This raises a critical question: beyond a simple network output, can we derive a formulation that more effectively captures these patterns and aligns with the goals of SISR?

On the other hand, image compression serves as a fundamental task in low-level vision applications, where the traditional compression standards Joint Video Experts Team (JVET) (2023); Wallace (1992); Taubman & Marcellin (2013) lay the ground work. The emerging learned image compression models Chamain et al. (2021); Ballé et al. (2018a); Guo-Hua et al. (2023); Liu et al. (2023a); Minnen et al. (2018); Cheng et al. (2020), compression algorithms of which mostly follow the pixel-space transform coding Chamain et al. (2021); Goyal (2001)paradigm, then introduce neural networks to further optimize compression efficiency by learning more compact latent representations and improving reconstruction quality. Specifically, they convert pixels into compact representations through a transform module, which eliminates the redundancy to reduce the bit cost in the subsequent entropy coding process. However, the core challenge of image compression is to accurately reconstruct the

information lost during compression and quantization. In other words, the models are to recover the impaired components in images, which can essentially be viewed as reconstructing a high-quality image from its 'low-resolution' version, much like Super-Resolution.

Based on the aforementioned analysis, it becomes clear that although Super-Resolution and Image Compression appear to be two distinct tasks, they share mutual similarities in two key aspects: (1) Both tasks require models to restore fine details from low-quality image content, as well as implicitly capture and reconstruct repetitive structural elements. (2) Both aim to conserve image quality, either by enhancing resolution or efficiently compressing data without significant loss of perceptual fidelity.

Hence, inspired by recent advances in decomposition fields and matrices factorization in 3D scene modeling Chen et al. (2022a); Müller et al. (2022); Chan et al. (2021); Fridovich-Keil et al. (2023); Cao & Johnson (2023); Chen et al. (2023a); Gao et al. (2023), we propose a unified representation, **Factorized Fields**, with generalizable Coefficient Backbone and Basis Transformer. This approach explicitly captures multi-scale visual features and repetitive structural components in images through a basis-coefficient decomposition. The resulting representation strikes a balance between being compact and information-rich, enabling the resolution of structural ambiguities and the precise modeling of image details through a multi-frequency formulation. In the meantime, such a formulation imposes a factorization constraint during model training, which not only enhances the quality of single-image super-resolution (SISR) but also reduces distortion and improves compression efficiency by explicitly modeling structural elements. On top of these, to leverage the robust information-recovering capability of SR models, we integrate such priors with Image Compression models, whose knowledge of detail compensation further refines the lost key elements.

Finally, we propose merging the bases by introducing an additional compression branch, consolidating multiple bases into one alongside multi-image transmission. This approach leverages the mutual information across multiple images, reducing the need for redundant transmissions and refining the basis structure. The main contributions of this paper are summarized as follows:

- We propose **Factorized Fields**, a unified representation that explicitly models multi-scale visual features and structural components for both super-resolution and image compression.

- We integrate super-resolution with image compression by introducing SR prior during decompression to compensate for lost details and developing a merged-basis compression branch for multi-image compression.

- We demonstrate state-of-the-art performance on benchmarks for both super-resolution and image compression through extensive experiments.

## 2 RELATED WORKS

**Super-Resolution (SR).** Image super-resolution is critical in computer vision, focusing on recovering high-resolution (HR) images from low-resolution (LR) inputs. Following the foundational studies, CNN-based strategies Dong et al. (2014)Lim et al. (2017b)Zhou et al. (2020)Sun et al. (2022)Kim et al. (2016) were initially introduced with modeling techniques such as residual learning Ledig et al. (2017b)Liu et al. (2020)Tong et al. (2017)Zhang et al. (2018b)Zhang et al. (2018a)Chih-Chung Hsu (2023)Lim et al. (2017a) , or recursive learning Chen et al. (2024)Tai et al. (2017). Besides, subsequent research also sheds light on GAN-base methodsLedig et al. (2017a)Ledig et al. (2017b)Wang et al. (2019)Wang et al. (2021) to enhance realisticity and detail quality. However, the inductive bias of CNN-based networks by restricting spatial locality hinders the capture of long-range dependency from images, which is alleviated by Transformer-based SISR networks Chen et al. (2021)Li et al. (2021). Afterward, SwinIRLiang et al. (2021) is proposed to combine spatial locality and non-local information by Swin TransformerLiu et al. (2021) with window attention and achieve breakthrough improvement in SISR. Following SwinIR's success, several works have built upon its frameworkConde et al. (2022)Zhu et al. (2023)Zhang et al. (2024a) to reach better image quality as well as solve information bottleneckChih-Chung Hsu (2023). Hybrid approaches CRAFT (Li et al., 2023) merge the benefits of convolutional and transformer structures to further elevate SR performance. For better feature aggregation, DAT (Chen et al., 2023d) and HAT (Chen et al., 2023b) integrate spatial and channel information using attention mechanisms to enhance their representation capabilities. Moreover, RGT (Chen et al., 2024) introduces a unique recursive-generalization self-attention mechanism that efficiently captures comprehensive spatial details with a linear increase in computational complexity.

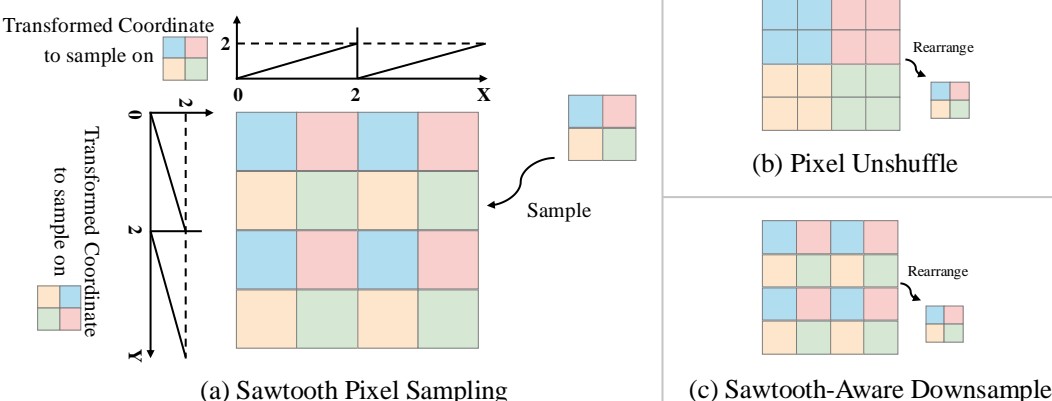

Figure 1: **The correlation between coordinate transformation and downsampling.** (a) The sawtooth transformation example with $k = 2$. (b) The PixelUnShuffle downsample. (c) To explicitly model the information for sampling with a sawtooth, we rearrange the feature map in a dilation-like manner in the downsample layer of the Basis Swin Transformer. This way, the feature sampled would capture the information in the original layout correctly.

**Image Compression (IC).** Deep learning has significantly advanced image compression, offering superior compression ratios and image quality compared to traditional methods like JPEG Wallace (1992) and JPEG-2000 Taubman & Marcellin (2013). Early CNN-based approaches Chamain et al. (2021); Ballé et al. (2018b); Guo et al. (2022) have been surpassed by transformer-based models Koyuncu et al. (2022); Zhu et al. (2022); Wu et al. (2021); Duan et al. (2023), which leverage spatio-channel attention for better performance. GAN-based methods Mao et al. (2023a;b); Feng et al. (2021); Rippel & Bourdev (2017) have further contributed to real-time adaptive compression. Recently, ELIC He et al. (2022) introduced efficient compression with unevenly grouped space-channel contextual adaptive coding, while LIC TCM Liu et al. (2023a) integrated transformers and CNNs to capture both global and local image features. The eContextformer Koyuncu et al. (2024) introduced patch-wise, checkered, and channel-wise grouping techniques for parallel context modeling with a shifted window spatial-channel attention mechanism. GroupedMixer Li et al. (2024) proposed a transformer-based entropy model with token mixers for inner and cross-group context modeling. Meanwhile, Wavelet Conditional Diffusion Song et al. (2024) introduced a wavelet-based model with uncertainty-aware loss, balancing high perceptual quality with low distortion.

In summary, while recent super-resolution and image compression advances focus on increasingly complex architectures, our work takes a different approach. We propose Factorized Fields, a unified framework that models visual and structural features, offering a more comprehensive solution to enhance performance in both tasks.

## 3 METHODS

In this section, we first briefly introduce the background of the factor fields Chen et al. (2023a), their properties, and also the preliminary of Learned Image Compression in Sec. 3.1. Inspired by factor fields, we next explain the motivation and how we derive our **Factorized Fields** for enhanced image reconstruction quality in Sec. 3.2. We then describe how to adapt the formulation to Super-Resolution in Sec. 3.3. Finally, we show how to incorporate such representation with image(s) compression and how we integrate Super-Resolution and Image Compression in Sec. 3.4.

### 3.1 PRELIMINARIES

**Factor Fields.** The concept of decomposition fields or factorized matrices in the reconstruction of 2D images or 3D scenes has shown superior rendering quality and improved efficiency recently Chen et al. (2022a); Gao et al. (2023); Müller et al. (2022); Fridovich-Keil et al. (2023); Cao & Johnson (2023); Chan et al. (2021), while Chen et al. (2023a) first proposed a unified representation called factor fields. For a 1D signal $s(x)$, the factor fields formulation is:

$$\hat{s}(x) = c(x)^T b(\gamma(x)), \tag{1}$$

where $c(x) = (c_1(x), ..., c_N(x))^T$ are spatial-varying coefficient fields, $b(x) = (b_1(x), ..., b_N(x))^T$ are basis functions, and $\gamma(x)$ is a coordinate transformation. The coefficient and basis are queried by sampling with the coordinates $x$. This generalizes to Q-dimensional signals:

$$\hat{s}(x) = \mathcal{P}(c(x) \odot b(\gamma(x))), \tag{2}$$

where $\mathcal{P}$ is a projection function and $\odot$ denotes the element-wise product. In practice, for multi-dimensional coefficients:

$$\hat{s}(x) = \mathcal{P}\Big(\mathbf{Concat}_{i=1}^N \Big\{ c_i(x) \odot b_i(\gamma_i(x)) \Big\}\Big). \tag{3}$$

This formulation allows the same basis to be applied at multiple spatial locations while varying the coefficients, especially when $\gamma$ is a periodic function.

**Learned Image Compression.**   Following (Minnen & Singh, 2020; Ballé et al., 2018a), a learned image compression model with a channel-wise entropy model can be formulated as:

$$
\begin{aligned}
z &= h_a(y; \phi_h), \ y = g_a(x; \phi), \\
\{X_{\text{mean}}, X_{\text{scale}}\} &= h_s(\hat{z}; \theta_h), \ \hat{z} = Q(z) \\
\hat{y} &= \{Q(y_0 - \mu_0) + \mu_0, ..., Q(y_t - \mu_t) + \mu_t\}, \ 0 <= t < l, \ \mu_t = e_i(\hat{y}_{<i}, X_{\text{mean}})) \\
\hat{x} &= g_s(\overline{y}; \theta), \ \overline{y} = \mathbf{Refine}_{\theta_r}(\mu_0, ..., \mu_t, \hat{y}).
\end{aligned}
\tag{4}
$$

The encoder $g_a$ transforms the raw image $x$ into a latent representation $y$. A hyper-prior encoder $h_a$ further processes $y$ to output $z$, capturing spatial dependencies. $z$ is quantized to $\hat{z}$, which is decoded by $h_s$ to produce features $X_{\text{mean}}$ and $X_{\text{scale}}$, used to estimate the mean $\mu$ and variance $\sigma$ of $y$. The latent $y$ is divided into $l$ slices, and each quantized around computed means $\mu_t$. These means are derived from earlier quantized slices and $X_{\text{mean}}$ by a slice network $e_i$. The quantized slices form $\hat{y}$. For decompression, $\hat{y}$ is refined using $\mathbf{Refine}_{\theta_r}$ based on $\mu_t$ and $\hat{y}$ to produce $\overline{y}$, approximating the original $y$. Finally, $g_s$ reconstructs the decompressed image $\hat{x}$ from $\overline{y}$. The model is trained using a Lagrangian multiplier-based rate-distortion optimization:

$$L = R(\hat{y}) + R(\hat{z}) + \lambda \cdot D(x, \hat{x}), \tag{5}$$

where $R(\hat{y})$ and $R(\hat{z})$ denote bit rates, $D(x, \hat{x})$ is the distortion term (calculated by MSE), and $\lambda$ balances compression efficiency and image fidelity. In our experiments, we follow Liu et al. (2023a), modifying only $g_s$ to demonstrate our representation's effectiveness.

## 3.2   FORMULATION OF FACTORIZED FIELDS

As discussed in Sec. 1, the key to superior rendering quality in image regression tasks of both Image Compression and Super-Resolution lies in the capability of the representation to capture accurate structural distribution and fine visual details. Meanwhile, transformations such as Fourier Transform or Wavelet Transform have long been used to model multi-frequency information Fuoli et al. (2021); Korkmaz et al. (2024) to express different implicit functions in images; however, such methods often suffer from under-expression due to the pre-defined and limited frequency bands or restriction from its formulation. Thus, we seek a representation that explicitly incorporates and fits multi-scale and multi-frequency components and yet is highly flexible and learnable according to individual features in an image.

Inspired from recent success on such decomposition fields Fridovich-Keil et al. (2023); Chen et al. (2023a; 2022a); Müller et al. (2022); Cao & Johnson (2023); Chan et al. (2021), we primarily built our **Factorized Fields** framework on factor fields Chen et al. (2023a) from Eq. 3:

$$\hat{I}(x) = \mathcal{P}\Big(\mathbf{Concat}_{i=1}^N \Big\{ c_i(x) \odot b_i(\gamma_i(x)) \Big\}\Big). \tag{6}$$

Note that $\hat{I}$ denotes the approximated images, and $x \in R^2$ are the pixel coordinates.

Such formulation has several key properties: First of all, by decomposing the images into basis frequencies, we can learn the implicit functions of an image and capture the mutual dependencies between pixels and across spatial composition; meanwhile, since the basis and coefficient are specific to every single image and both learnable in all spatial dimensions, the restriction on a limited number of basis (we use $N = 6$ in all of our experiments) can be alleviated. Finally, as in Chen et al.

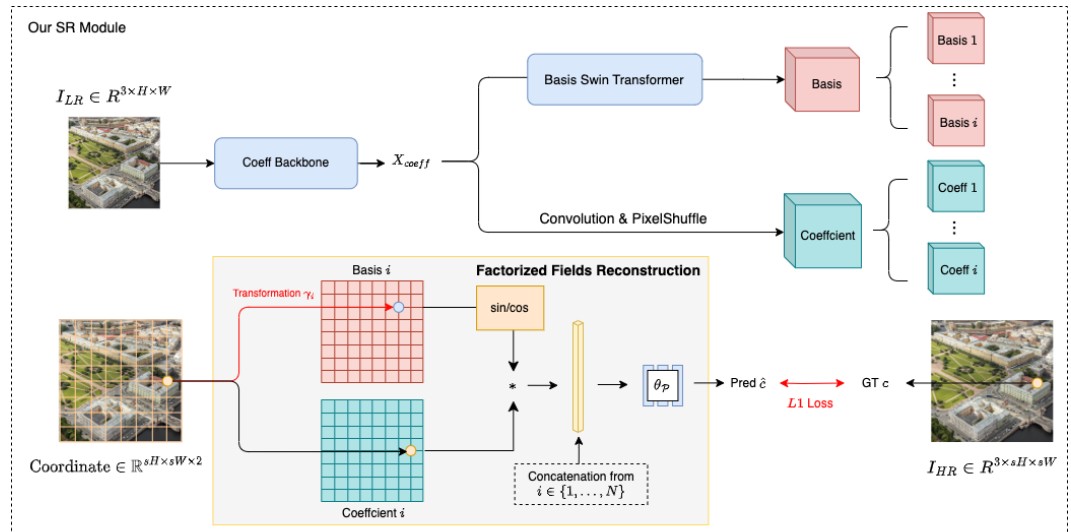

Figure 2: **The overall pipeline of image super-resolution with our Factorized Fields.** Given a low-resolution image $I_{LR}$, we first extract coefficient feature map $X_{\text{coeff}}$ with the coefficient backbone, which is then decoded into coefficient and passed through the basis Swin Transformer for basis, separately. Finally, as in Eq. 8, the coefficient and basis are sampled, multiplied, and decoded for final high-resolution output $I_{HR}$, where $s$, $H$, $W$ denote the scale factor, height, and width respectively.

(2023a), of all the tested coordinate transformation $\gamma$ in Eq. 3 sawtooth $\gamma(x) = x \bmod k, k \in \mathbb{R}$ performs mostly the best in image regression tasks. We can easily observe that such transformation implicitly captures patch-like frequency information as shown in Fig. 1a, and thus, we propose that by leveraging the inter-patch information from the sawtooth coordinate transformation, the visual correspondence between spatial locations can be effectively represented.

However, in practice, we sample the basis via bilinear or bicubic sampling due to memory constraints, *i.e.*, the feature size is less than image height and width, and this poses a severe problem: These nearby sampled pixel features are actually linear or based on cubic interpolation with respect to the basis, and such inductive bias hinders the representation of non-linearity in images. To resolve this, we propose a modified version, our Factorized Fields:

$$\hat{I}(x) = \mathcal{P}\Big(\textbf{Concat}_{i=1}^{N}\,_{j=1}^{K}\Big\{c_i(x) \odot \psi(\alpha_j \cdot b_i(\gamma_i(x)))\Big\}\Big), \psi \in \{\sin, \cos\}, \alpha_j \in R. \tag{7}$$

Here, inspired by the Fourier Series, with a scalar $\alpha$ which can be viewed as increasing to sampling frequency and a transformation function $\psi$ to add to implicit non-linearity functions and modulation of product, our complete Factorized Fields method stands out by the fact that such formulation significantly enhances the ability to represent complex non-linear structures in images and effectively composes high-frequency components between pixels.

### 3.3 SUPER-RESOLUTION WITH FACTORIZED FIELDS

We represent a super-resolved image using our Factorized Fields, where coefficients and basis are generated by networks $F_{\text{coeff}}$ and $F_{\text{basis}}$ from a low-resolution Image:

$$\hat{I}_{\text{SR}}(x) = \mathcal{P}\Big(\textbf{Concat}_{i=1}^{N}\,_{j=1}^{K}\Big\{c_i^{\text{LR}}(x) \odot \psi(\alpha_j \cdot b_i^{\text{LR}}(\gamma_i(x)))\Big\}\Big), \tag{8}$$

where $c_i^{\text{LR}}(x) = \text{Conv}(F_{\textbf{coeff}}(I_{\text{LR}}))_i(x)$ and $b_i^{\text{LR}}(x) = F_{\textbf{basis}}(F_{\textbf{coeff}}(I_{\text{LR}}))_i(\gamma_i(x))$. Note that we sample the outputs $\text{Conv}(F_{\textbf{coeff}}(I_{\text{LR}}))$ and $F_{\textbf{basis}}(F_{\textbf{coeff}}(I_{\text{LR}}))$ with coordinates $x$ and $\gamma(x)$, respectively.

Our model comprises three main components: Coefficient Backbone, Basis Swin Transformer, and Factorized Fields Reconstruction. As shown in Fig. 2, the process begins with $I_{LR} \in \mathbb{R}^{3 \times H \times W}$. The Coefficient Backbone extracts features $X_{\text{coeff}} \in \mathbb{R}^{C_c \times H_c \times W_c}$, which are then used to generate coefficients $c$ through convolution and pixel shuffle operations, and fed into the Basis Swin Transformer to produce a multi-scale basis $b = \{b_1, ..., b_N\}$, $b_i \in \mathbb{R}^{C_{b_i} \times H_{b_i} \times W_{b_i}}$. The coefficients and basis are combined to reconstruct $I_{SR} \in \mathbb{R}^{3 \times sH \times sW}$ using Eq. 8, where $s$ is the scale factor. In our

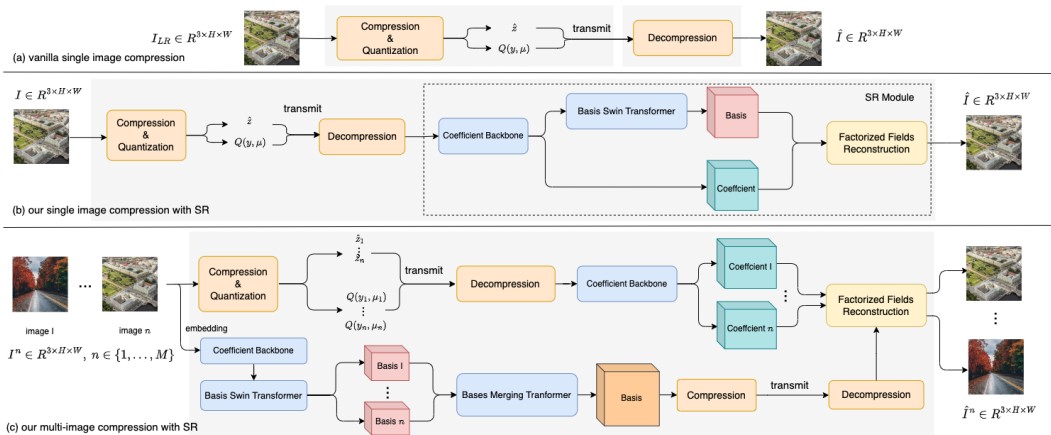

Figure 3: **The illustration of our joint image-compression and super-resolution framework compared with the traditional compression-only method.** (a) Traditional learning-based compression methods (b) Our approach surpasses (a) by incorporating our Super-Resolution (SR) Module from Sec.3.3 as information-recovery prior, detailed in Sec. 3.4.1. (c) Expanding on (b), Sec. 3.4.2 introduces a multi-image compression strategy that utilizes both our SR Module and a Basis Merging Transformer to capture shared structure.

experiments, $\gamma_i(x)$ is a Sawtooth function: $\gamma_i(x) = x \mod k_i$, with $k_i \in R$. We optimize model parameters using an $L_1$ loss function. To demonstrate our method's effectiveness, we use existing SR methods Chen et al. (2023c;d); Zhang et al. (2024a) as the Coefficient Backbone. For the Basis Swin Transformer, we employ Swin Transformer BlocksLiu et al. (2022) with a series of downsampling operations. We use a dilation-like downsampling technique (Fig. 1(c)) to accommodate the sawtooth sampling pattern. The final basis is refined using additional upsampling and convolution layers.

### 3.4 IMAGE COMPRESSION WITH FACTORIZED FIELDS AND SUPER-RESOLUTION

Image Compression, at its core, is to strike a balance between the amount of information contained in the latent bits and the final image quality. However, as discussed in Sec. 1, 2, and 3.2, most recent works draw emphasis on how to retrieve implicit elements through designing different architectures, such as analysis transforms and entropy models Koyuncu et al. (2024); Li et al. (2024), or decompression modules Song et al. (2024); Duan et al. (2023), while this paper aims at addressing the representation itself to better capture the structural correlations and thus acheive better image quality through explicit modeling of freqeuncy components and Factorized Fields formulation as in Eq. 7. In addition, with our trained SR model described in Sec. 3.3, it intuitively serves as a strong prior for information recovery, i.e., it contains extensive knowledge of how to reconstruct missing details and enhance image quality by leveraging learned patterns from the training data. Thus, since Super-Resolution and Image Compression share the core principle of reconstructing and enhancing image details from low-quality sources, we can effectively integrate this prior into the compression pipeline. In the following, we respectively present the proposed joint framework for the single and multi-image compression tasks.

### 3.4.1 SINGLE IMAGE COMPRESSION

The overall pipeline is shown in Fig. 3(b). To demonstrate the robustness of our representation and the effectiveness of the SR prior, the compression and decompression networks greatly follow Liu et al. (2023a), with only the synthesis transform replaced by our SR pipeline, where the details can be referenced in Supplementary Materials. In practice, the training is performed in two stages. After we obtain the trained SR prior, the model is fine-tuned with a lower learning rate alongside the compression module, which is then trained end-to-end with the loss function defined in Eq. 5.

### 3.4.2 MULTI-IMAGE COMPRESSION

For each basis $b_i \in R^{C_{b_i} \times H_{b_i} \times W_{b_i}}$ associated with any arbitrary image, we can consider it as encapsulating the inherent pixel structure. These bases can be combined into a unified generic basis that captures the structural distribution of images and potentially reduces noise. Given $M$ images

Table 1: **Quantitative comparisons on** $4\times$ **super-resolution with state-of-the-art methods.** The best results are colored red. The models with † are those who use same-task pretraining Chen et al. (2023c). Please refer to **quantitative results** in Sec. 4.1 for details.

| Method | Params (M) | MACs (G) | Forward Pass Memory (MB) | Set5 PSNR↑ | Set5 SSIM↑ | Set14 PSNR↑ | Set14 SSIM↑ | B100 PSNR↑ | B100 SSIM↑ | Urban100 PSNR↑ | Urban100 SSIM↑ | Manga109 PSNR↑ | Manga109 SSIM↑ |
|---|---|---|---|---|---|---|---|---|---|---|---|---|---|
| SwinIR (Liang et al., 2021)(ICCV2021W) | 28.01 | 119.68 | 3,826 | 32.93 | 0.9043 | 29.15 | 0.7958 | 27.95 | 0.7484 | 27.56 | 0.8273 | 32.22 | 0.9273 |
| ATD (Zhang et al., 2024b)(CVPR2024) | 20.26 | 77.10 | 6,572 | 33.14 | 0.9061 | 29.25 | 0.7976 | 28.02 | 0.7524 | 28.22 | 0.8414 | 32.65 | 0.9308 |
| DAT (Chen et al., 2023d))(ICCV2023) | 14.80 | 61.66 | 4,192 | 33.15 | 0.9062 | 29.29 | 0.7983 | 28.03 | 0.7518 | 27.99 | 0.8365 | 32.67 | 0.9301 |
| RGT (Chen et al., 2024)(ICLR2024) | 13.37 | 834.25 | 3,404 | 33.16 | 0.9066 | 29.28 | 0.7979 | 28.03 | 0.7520 | 28.09 | 0.8388 | 32.68 | 0.9303 |
| HAT† (Chen et al., 2023b)(CVPR2023) | 20.77 | 86.02 | 3,692 | 33.18 | 0.9073 | 29.38 | 0.8001 | 28.05 | 0.7534 | 28.37 | 0.8447 | 32.87 | 0.9319 |
| HAT-L† (Chen et al., 2023b)(CVPR2023) | 40.84 | 167.27 | 6,804 | 33.30 | 0.9083 | 29.47 | 0.8015 | 28.09 | 0.7551 | 28.60 | 0.8498 | 33.09 | 0.9335 |
| ATD-L* | 49.42 | 184.83 | 15,582 | 33.12 | 0.9062 | 29.31 | 0.7985 | 28.02 | 0.7514 | 28.25 | 0.8422 | 32.78 | 0.9309 |
| DAT-L* | 43.01 | 175.42 | 11,326 | 33.33 | 0.9084 | 29.40 | 0.8009 | 28.04 | 0.7543 | 28.49 | 0.8473 | 33.02 | 0.9321 |
| ATD-F† (Ours) | 45.46 | 149.87 | 8,674 | 33.29 | 0.9082 | 29.48 | 0.8017 | 28.03 | 0.7539 | 28.53 | 0.8487 | 33.11 | 0.9335 |
| DAT-F† (Ours) | 40.00 | 134.42 | 6,206 | 33.45 | 0.9094 | 29.60 | 0.8039 | 28.13 | 0.7560 | 28.75 | 0.8520 | 33.23 | 0.9339 |
| HAT-F† (Ours) | 45.97 | 158.79 | 5,750 | 33.53 | 0.9100 | 29.65 | 0.8050 | 28.18 | 0.7569 | 28.79 | 0.8527 | 33.33 | 0.9342 |
| HAT-F-ImageNet† (Ours) | 45.97 | 158.79 | 5,750 | 33.55 | 0.9102 | 29.63 | 0.8049 | 28.18 | 0.7569 | 28.80 | 0.8529 | 33.33 | 0.9342 |
| HAT-L-F† (Ours) | 66.04 | 240.03 | 8,888 | 33.75 | 0.9116 | 29.87 | 0.8091 | 28.31 | 0.7597 | 29.51 | 0.8637 | 33.36 | 0.9343 |
| HAT-F-Basis-First† (Ours) | 46.67 | 161.66 | 5,696 | 33.33 | 0.9085 | 29.47 | 0.8015 | 28.10 | 0.7554 | 28.57 | 0.8494 | 33.14 | 0.9336 |
| HAT-F-Concat† (Ours) | 45.52 | 129.05 | 4,826 | 33.46 | 0.9095 | 29.57 | 0.8035 | 28.16 | 0.7566 | 28.73 | 0.8518 | 33.28 | 0.9341 |

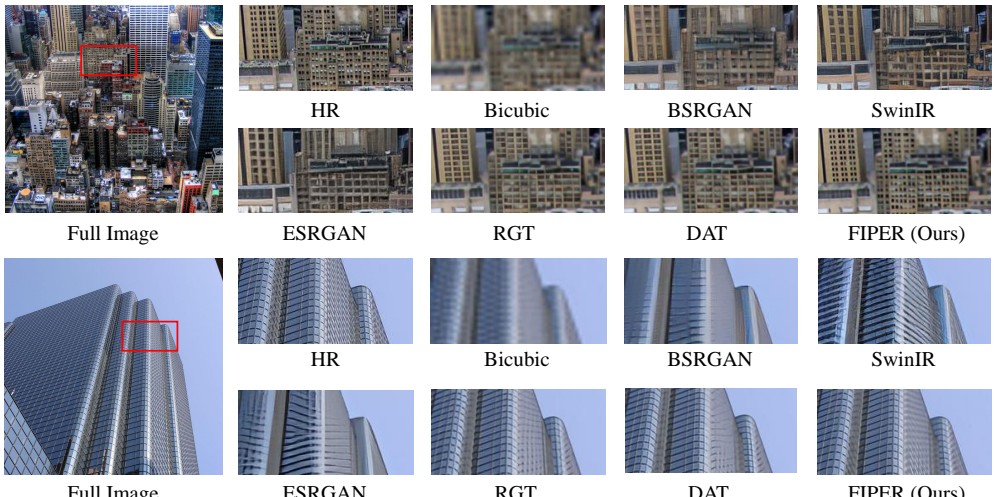

Figure 4: **Visual comparisons on super-resolution (4×).**

and their respective bases $b_i^n, n \in \{1, ..., M\}$, we apply a Basis Merging Transformer $F_{\text{merge}}$ at each location to integrate the $M$ elements:

$$b_i(h, w) = F_{\text{merge}}\left(\{b_i^n(h, w) \mid n \in 1, ..., M\}\right), \ 0 \le h < H_{b_i}, \ 0 \le w < W_{b_i}. \quad (9)$$

Here, $F_{\text{merge}}$ is a standard transformer, similar in architecture to that described in (Oquab et al., 2023). We treat the bases as tokens and prepend the sequence with a $CLS$ token, concluding the sequence to form the final merged basis.

Since compression induces error, according to Fig. 3b, the Coefficient feature map $X_{\text{coeff}}$ generated with Coefficient Backbone contains misinformation, and Basis Swin Transformer that uses such features would capture the information wrongfully for the basis and thus amplify error. To solve this, we exploit the mergeable property of basis, compress and transmit the merged basis independently along other quantized variables $\{Q(y, \mu), \ \hat{z}\}$, and finally reconstruct the $M$ images with the merged basis and their individual decoded coefficients. as illustrated in Fig. 3c. This way, we can enjoy less information loss with basis while maintaining low bit rates.

## 4 EXPERIMENTS

### 4.1 IMAGE SUPER-RESOLUTION

**Experimental setup.** We conduct extensive experiments to validate the effectiveness of our Factorized Fields representation for Super-Resolution tasks. Following the strategy outlined in (Chen et al., 2023b) and (Chih-Chung Hsu, 2023), we adapt the same-task pretraining approach for all the Super-Resolution models. Unlike these previous works, we leverage the SA-1B dataset from

(Kirillov et al., 2023), which includes approximately 10 million images, where we randomly sample 4 million for training, which is much less that of ImageNet (Deng et al., 2009), which is used for Chen et al. (2023b) and Chih-Chung Hsu (2023) pretraining. Note that we use SA-1B just for its content-rich and high-resolution images since SA-1B is designed for training segmentation models, whereas ImageNet focuses primarily on single-class prediction. We further conduct experiments to compare training performance with SA-1B and ImageNet in Tab. 1. While the performances are mostly equivalent, we witness faster convergence with SA-1B during training.

To be more focused on the representation itself, we utilize various pre-trained SR models (SwinIR (Liang et al., 2021), HAT (Chen et al., 2023c), DAT (Chen et al., 2023d), and ATD (Zhang et al., 2024b)) as the Coefficient Backbone, with a consistent Basis Transformer architecture. Training involves initializing the Coefficient Backbones from pre-trained SR models and randomly initializing the Basis Transformers. Models are pre-trained for 300k iterations on the SA-1B dataset. After pretraining, we use DF2K (DIV2K (Agustsson & Timofte, 2017) + Flickr2K (Lim et al., 2017b)) as the finetuning dataset following (Conde et al., 2022; Chen et al., 2023b) for 200k iterations.

For pretraining, we utilize AdamW optimizer with learning rate 1e-4, batch size 16, betas (0.9, 0.99), and other parameters set to PyTorch default, while we use learning rate 1e-5 during finetuning. Throughout training, the input is randomly cropped to $256 \times 256$ and bicubically resized to $64 \times 64$ for Coefficient Backbone input. As for the hyperparameter of Factorized Fields, the number $N$ of coefficient and basis is set to 6, and the scalars $\alpha_j$ are set to $\{1, 4, 16, 64\}$ since we want to capture both base frequency $\alpha_j = 1$ and high frequency $\alpha_j = 64$ information.

**Quantitative results.** Tab. 1 presents the quantitative comparison between our approach and state-of-the-art (SoTA) methods. We evaluate the methods using five benchmark datasets, including Set5 (Bevilacqua et al., 2012), Set14 (Zeyde et al., 2010), BSD100 (Martin et al., 2001), Urban100 (Huang et al., 2015), and Manga109 (Matsui et al., 2017). For quantitative metrics, PSNR and SSIM are reported. The average relative improvement of **204.4%** in PSNR over the baseline across the five datasets is calculated using the formula $(c - b)/(a - b)$, where $b$ represents the PSNR of a representative baseline, SwinIR, $a$ represents the PSNR of SOTA, HAT-L, and $c$ represents the PSNR of our best model, HAT-L-F. This formula measures the relative performance gain of our model compared to the gap between HAT-L and SwinIR.

To validate the effectiveness of our framework, we employ three different SoTA SR models—ATD Zhang et al. (2024a), DAT Chen et al. (2023d), and HAT Chen et al. (2023c)—as the Coefficient Backbone in our pipeline, which we denote as ATD-F, DAT-F, and HAT-F, respectively. These models exhibit significant improvements when compared to their counterparts ATD-L, DAT-L, and HAT-L, which possess similar parameter counts. It is important to note that only Chen et al. (2023c) provides a large-scale model. To maintain fairness, we scale up ATD and DAT to match this size and train them under the same configuration as our model, including both pretraining and fine-tuning stages. To ensure an equitable comparison with other methods, we further train another model, HAT-F-ImageNet, using ImageNet as the pretraining dataset, following the protocols outlined in Chen et al. (2021; 2023c). The results demonstrate that its performance remains consistent with only minor perturbations.

Furthermore, in traditional Fourier Series and other image processing methods Wallace (1992), the basis is typically derived first and then used to compute the coefficients. In contrast, our method derives the coefficient features first, as illustrated in Fig. 2. To explore this difference, we develop another variant of our model, denoted HAT-F-Basis-First, where we reverse the order of operations. In this case, we first pass the image through the Basis Swin Transformer and then use the resulting basis features and the image input to derive the coefficients. This approach, however, leads to a gigantic performance drop, showing the importance of the order of the pipeline. Specifically, we argue that in our pipeline, the Coefficient Backbone functions more as a feature extraction module, where the refined features facilitate downstream basis extraction.

Lastly, to evaluate the effectiveness of our Factorized Fields, we trained a model named HAT-F-Concat, which does not apply the formulation in Eq. 7. Instead, it concatenates the basis and coefficient directly and decodes the resulting features to produce the output. Although this approach results in reduced performance, which indicates the representation does act as an imperative role in modeling image information, the Basis Swin Transformer with Sawtooth downsampling still

Table 2: **Comprehensive evaluation for image compression.** Using VTM as an anchor for calculating BD-Rate. Latencies are measured under an NVIDIA GTX 3090 GPU.

| Method | BD-Rate (%) ↓ | Latency(s) | | Params(M) |
| --- | --- | --- | --- | --- |
| | | Tot Enc ↓ | Tot Dec ↓ | |
| VTM | 0.00 | 129.21 | 0.14 | - |
| Xie (MM 21') | -0.78 | 2.93 | 6.00 | 50.0 |
| Cheng (CVPR 22') | 5.44 | 1.98 | 4.69 | 29.6 |
| STF (CVPR 22') | -4.31 | 0.14 | 0.13 | 99.9 |
| ELIC (CVPR 22') | -7.24 | 0.07 | 0.09 | 36.9 |
| TCM (CVPR 23') | -11.74 | 0.16 | 0.15 | 76.7 |
| TCM-HAT-L-F (Ours) | -21.09 | 0.109 | 0.264 | 110.34 |
| TCM-HAT-F-multi M=1 (Ours) | 27.96 | 0.232 | 0.174 | 131.35 |
| TCM-HAT-F-multi M=2 (Ours) | 2.70 | 0.232 | 0.174 | 131.35 |
| TCM-HAT-F-multi M=4 (Ours) | -10.11 | 0.232 | 0.174 | 131.35 |
| TCM-HAT-F-multi M=8 (Ours) | -16.61 | 0.232 | 0.174 | 131.35 |
| TCM-HAT-F-multi M=16 (Ours) | -19.88 | 0.232 | 0.174 | 131.35 |
| TCM-HAT-F-multi M=24 (Ours) | -20.97 | 0.232 | 0.174 | 131.35 |

(a) Kodak (b) CLIC (c) Tecnick

Figure 5: **Performance (RD-Curve) evaluation on image compression using different datasets.**

contributes to improved reconstruction, even without Factorized-Fields decoding, highlighting its effectiveness.

**Visual comparison.** We provide the visual comparison in Fig. 4. The images are randomly sampled from the DIV2K dataset. Our method faithfully reconstructs the image details, whereas the other approaches suffer from over-smoothing or hallucinating details absent in the ground truth.

## 4.2 SINGLE- AND MULTI-IMAGE COMPRESSIONS

**Experimental Setup** We evaluate our Factorized Fields representation for image compression tasks, comparing it against state-of-the-art methods. Following our Super-Resolution setting in Experiment Setup from Sec. 4.1, we use the same set of SA-1B for training. To emphasize our representation, we initialize compression and decompression modules in Fig.3 from pre-trained Liu et al. (2023a) and the SR Module from those in Sec. 4.1, we then train the pipeline end to end on 256x256 patches for 200k iterations, with AdamW optimizer (Loshchilov, 2017) with learning rate 1e-5, batch size 16, betas (0.9, 0.99) and other parameters set to PyTorch default.

We integrate our SR Module with the pre-trained compression module TCM, creating TCM-HAT-F and TCM-HAT-L-F models. TCM-HAT-F-multi represents the multi-image compression pipeline. For multi-image compression in Fig. 3(c), we set the Basis Swin Transformer and the Basis Merging Transformer to be trainable while the other parts remain frozen.

**Rate-Distortion Performance Comparison** We compare our model with State-of-the-Art learned end-to-end image compression algorithms, including (Liu et al., 2023b), (Chen et al., 2022b), Zou et al. (2022), Xie et al. (2021), Cheng et al. (2020), Ballé et al. (2018a), Li et al. (2024), Jiang et al. (2023), Minnen & Singh (2020), Bellard, Qi et al. (2023), and He et al. (2022). The classical image compression codec, VVC (Team, 2021), is also tested by using VTM12.1.The rate-distortion performance on various datasets, including Kodak, Tecnick old test set with resolution 1200×1200, and CLIC Professional Validation, is shown in Fig. 5.

Table 3: **Comparison of improvements of Factorized Fields.** $\psi$ and $\alpha$ are the same in Eq. 8

| Metric | PSNR ↑ | SSIM ↑ | LPIPS ↓ |
|---|---|---|---|
| Baseline (Chen et al., 2023a) | 22.04 | 0.505 | 0.5296 |
| Ours | 38.44 | 0.999 | 0.0385 |
| No $\psi$ to control magnitude | 13.46 | 0.147 | 0.766 |
| No $\alpha$ for pixel-wise frequency information | 21.25 | 0.537 | 0.527 |

Table 4: **Validation of the effectiveness of SR prior.** The best PSNR in marked in color red.

| Method | Kodak | | | | | | CLIC | | | | | | Tecnick | | | | | |
|---|---|---|---|---|---|---|---|---|---|---|---|---|---|---|---|---|---|---|
| | $\lambda = 0.0025$ | | $\lambda = 0.0067$ | | $\lambda = 0.025$ | | $\lambda = 0.0025$ | | $\lambda = 0.0067$ | | $\lambda = 0.025$ | | $\lambda = 0.0025$ | | $\lambda = 0.0067$ | | $\lambda = 0.025$ | |
| | bpp | PSNR↑ | bpp | PSNR↑ | bpp | PSNR↑ | bpp | PSNR↑ | bpp | PSNR↑ | bpp | PSNR↑ | bpp | PSNR↑ | bpp | PSNR↑ | bpp | PSNR↑ |
| TCMLiu et al. (2023a) | 0.1533 | 30.0834 | 0.2983 | 32.5841 | 0.6253 | 36.1345 | 0.1214 | 31.8207 | 0.2235 | 34.2098 | 0.4503 | 37.1201 | 0.1268 | 32.0588 | 0.2193 | 34.3669 | 0.3981 | 36.9066 |
| TCM-HAT-F-Scratch | 0.1570 | 30.0857 | 0.2976 | 32.5893 | 0.6211 | 36.1389 | 0.1214 | 31.9421 | 0.2235 | 34.2894 | 0.4503 | 37.1434 | 0.1258 | 32.0632 | 0.2189 | 34.3781 | 0.4001 | 36.9223 |
| TCM-HAT | 0.1567 | 30.1843 | 0.2992 | 32.6454 | 0.6268 | 36.2267 | 0.1220 | 31.9737 | 0.2266 | 34.3319 | 0.4512 | 37.2486 | 0.1262 | 32.1423 | 0.2174 | 34.5124 | 0.3971 | 36.9934 |
| TCM-HAT-F | 0.1574 | 30.4012 | 0.2998 | 32.8910 | 0.6276 | 36.4461 | 0.1229 | 32.1917 | 0.2249 | 34.4109 | 0.4512 | 37.3135 | 0.1255 | 32.4591 | 0.2186 | 34.7656 | 0.3975 | 37.3244 |

In Tab. 2, our TCM-HAT-L-F model achieved a significant BD-Rate improvement of -21.09% compared to VTM, outperforming previous state-of-the-art methods. The multi-image compression approach (TCM-HAT-F-multi) shows increasing performance gains with the number of images compressed simultaneously, reaching -20.97% BD-Rate improvement for $M = 24$. The result shows that direct transmission of bases would indeed reduce the error from Coefficient Backbone to Basis Swin Transformer and that the distortion increase with $M$, as the information contained in the merged basis is limited and merging multiple bases into one would cause increasing information loss.

Our analysis reveals several significant advantages of the FIPER framework in image compression tasks. The approach demonstrates substantial improvements in compression efficiency across various benchmark datasets, including Kodak, CLIC, and Tecnick, indicating its broad applicability. Notably, the multi-image compression strategy shows particularly promising results for larger image sets, suggesting scalability benefits. Furthermore, our method maintains competitive latency while significantly improving compression performance and balancing efficiency and quality.

### 4.3 ABLATION STUDIES

**Effectiveness of Factorized Fields design.** We conduct experiments to verify our modification of Factorized Fields, modified from Eq. 3 to Eq. 7. The quantitative performance reported on single-image regression is shown in Tab. 3, where each result is measured after 256 iterations. Compared to baseline results, our refinements in modeling pixel-level frequency have significantly improved all performance metrics. Additionally, our results demonstrate that the modulation function $\psi$ and the scalar $\alpha$ are interdependent, each essential to the other's function.

**Influence of SR priors in Image Compression** We conduct experiments with various configurations to verify the proposed image compression pipeline's effectiveness. As shown in Tab. 4, we present the quantitative performance of models trained with different values of $\lambda$ in Eq. 5. Specifically, TCM-HAT refers to substituting our SR Module with the original HAT Chen et al. (2023c) in the pipeline illustrated in Fig. 3.b. TCM-HAT-F represents our complete pipeline, while TCM-HAT-F-Scratch denotes the same pipeline but with the SR Module initialized randomly. Our results demonstrate that integrating SR priors with image compression improves performance, and our proposed representation further enhances results. This highlights the robustness of our Factorized Fields in capturing fine details in image regression tasks.

## 5 CONCLUSION

We proposed Factorized Fields, a representation that models implicit structures and patterns by decomposing images into multi-frequency components. This approach addresses challenges in Super-Resolution and Image Compression by restoring details and preserving visual fidelity. We integrate SR priors with Image Compression for improved information recovery and introduce a basis merging technique for enhanced rendering quality across multiple images. Experiments demonstrate state-of-the-art performance in both SISR and Image Compression benchmarks, addressing limitations of previous methods.

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

# A   ARCHITECTURE DETAILS

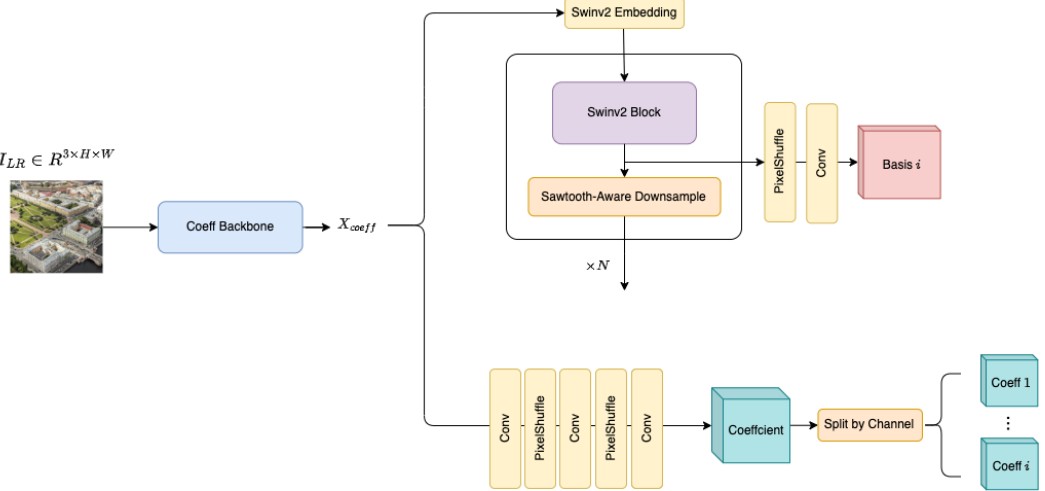

Figure 6: **Detailed architecture of Super-Resolution Modules**

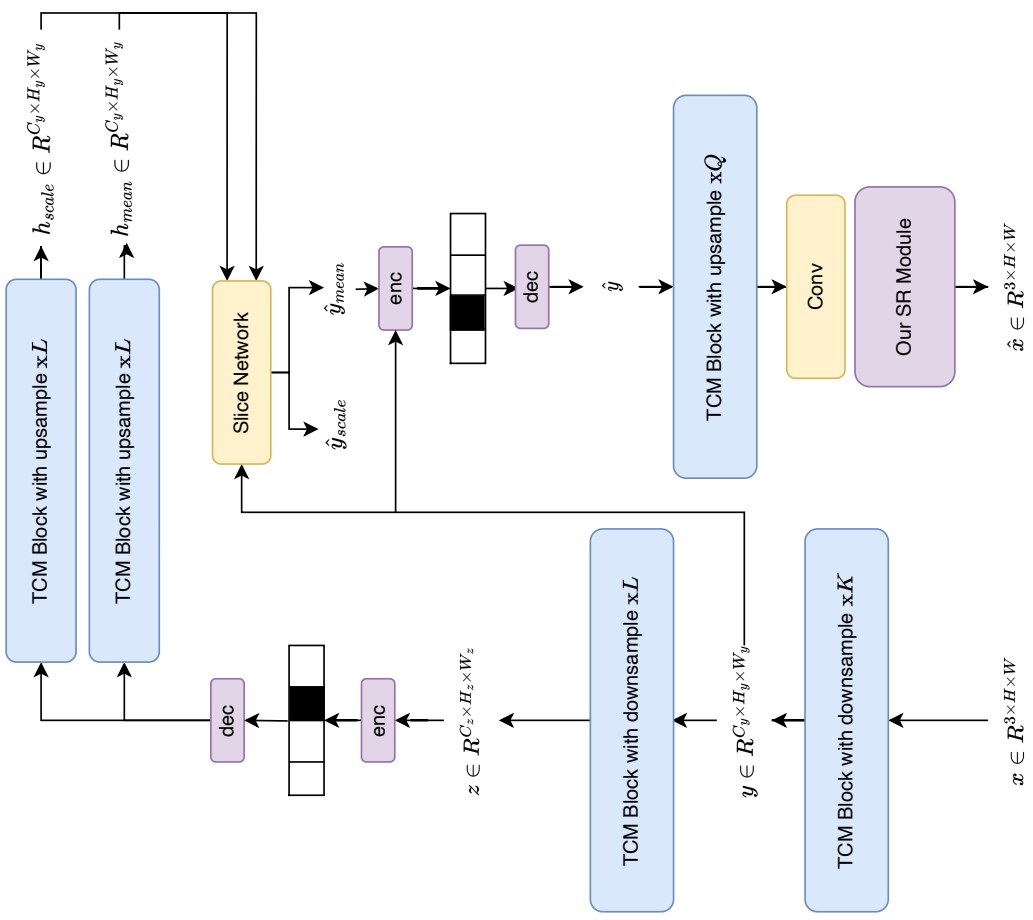

Figure 7: **Detailed architecture of Compression Pipeline in Fig.3(b)**.

**Super-Resolution**   The architecture of our Super-Resolution modules is shown in Fig.6. We extract the feature map $X_{coeff}$ of Coeff Backbone after the last layer, before upsampling. The hidden

dimension of Basis Swin Transformer is set to 384 with 8 heads in attention, and each blocks contains 2 Swinv2 Layers with window attention. The sawtooth-aware downsample reduces the height and width by half, where upsample scalar of Pixelshuffle for each basis output is set to 4.

**Image Compression** The architecture of our Image Compression pipeline of Fig.3(b) is shown in Fig.7. We extract intermediate features with height and width 128 and convolve with stride 2 for SR Module input.

## B VISUALIZATION

| HR | Bicubic | SwinIR | RGT | DAT | FIPER (ours) |
|----|---------|--------|-----|-----|--------------|

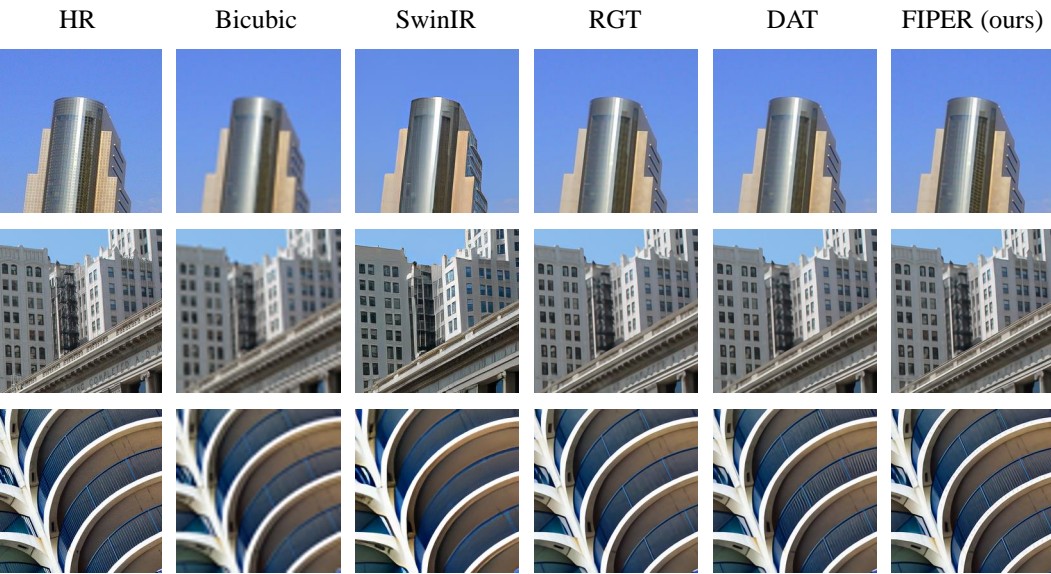

Figure 8: **Visual comparisons on super-resolution (4×).**

