# OpenReview forum: "FIPER: Generalizable Factorized Fields for Joint Image Compression and Super-Resolution"
_ICLR.cc/2025/Conference — ICLR 2025 Conference Withdrawn Submission_

### Official Review · Reviewer_FoRw · 2024-10-30

**Soundness:** 2
**Presentation:** 2
**Contribution:** 2
**Rating:** 3
**Confidence:** 4

**Summary:**

The paper proposes a Factorized Fields framework to addresses challenges in SR and Image Compression by restoring details and preserving visual fidelity. the author integrates SR priors with Image Compression for improved information recovery and introduce a basis merging technique for enhanced rendering quality across multiple images. Experiments demonstrate the proposed method obtains better results.

**Strengths:**

1. The paper proposes a unified representation, named Factorized Fields, for both image super-resolution and image compression.
2. The method explicitly captures multi-scale visual features and structural components in images through basis-coefficient decomposition.
3. The paper claims state-of-the-art performance in both Super-Resolution and Image Compression.

**Weaknesses:**

1. The Factorized Fields approach may be computationally complex, especially when dealing with large size data.
2. The paper presents a unified framework, but further validation is needed to see how the method performs under compression ratios.
3. Implementation and training details are not sufficiently detailed.

**Questions:**

1. What task was used in the ablation experiment and what dataset was used for validation?
2. The proposed Factorized Fields framework looks like integrating two different Transformer models to learn different components of an image, what would be the effect of replacing one of them with a CNN model?
3. Is this method effective in dealing with compressed image super-resolution?

---

### Official Review · Reviewer_TpAS · 2024-11-01

**Soundness:** 3
**Presentation:** 2
**Contribution:** 2
**Rating:** 5
**Confidence:** 4

**Summary:**

This paper proposes the use of Factorized Fields to address super-resolution and image compression (post-processing enhancement). Factorized Fields decompose signals into a set of bases and a set of covariances, utilizing these to restore the signals. Experiments demonstrate the effectiveness of Factorized Fields in both super-resolution and multi-image compression tasks.

**Strengths:**

The introduction of Factorized Fields to image super-resolution and multi-image compression is intriguing. I find that, based on the paper's experiments, the performance in image super-resolution is quite good, even though I am not an expert in SR. Additionally, Factorized Fields shows sufficient advantages when jointly compressing 8 images.

**Weaknesses:**

1. The Factorized Fields presented in this paper have already been introduced in reference [1], and there are no significant modifications made to Factorized Fields itself. While the paper introduces a Coefficient Backbone and Basis Swin Transformer for signal restoration based on covariances and bases, I believe the novelty is somewhat limited.


2. In single-image compression, the introduction of Factorized Fields actually leads to a performance drop. In single-image compression, there is no need to transmit the bases and covariances. Furthermore, when enhancing the decoded image with Factorized Fields, the performance is worse, which raises questions about the method's effectiveness.


3. For multi-image compression, the pipeline differs from that of single-image compression. Why not estimate the bases and covariances at the decoding stage for reconstruction?

[1] Chen, Anpei, et al. "Factor fields: A unified framework for neural fields and beyond." arXiv preprint arXiv:2302.01226 (2023).

**Questions:**

1. Why does the comparison in Figure 5 include MLIC only for the Tecnick dataset, while other datasets are not included?

2. Why is the encoding and decoding time exactly the same in Table 2, despite different values of $M$?

---

> ### Author Response · Authors · 2024-11-13
>
> Dear reviewer TpAS :
>
> The covariance term mentioned  in your question seems a bit confusing, would you mind elaborating on the concept you are referring to ?
>
> Thanks

---

### Official Review · Reviewer_eBXm · 2024-11-04

**Soundness:** 2
**Presentation:** 1
**Contribution:** 2
**Rating:** 5
**Confidence:** 5

**Summary:**

The paper focuses on the joint tasks of image compression and super-resolution. It proposes a unified representation, Factorized Fields, which is based on a basis coefficient decomposition to capture visual features and structural components. The authors develop an SR model with a Coefficient Backbone and a Basis Swin Transformer and integrate it with image compression. They also introduce a merged-basis compression branch and use the trained SR modules as priors in the compression pipeline.

**Strengths:**

Innovative Concept in SR Module: Introducing the Factorized Fields idea in the SR module is a significant contribution. This novel approach has the potential to represent image details better and could lead to improved performance in super-resolution tasks.

Performance Gains: The experimental results suggest that the proposed method achieves notable performance improvements. The reported gains in PSNR for super-resolution and BD-rate reduction for image compression are quite impressive, indicating the effectiveness of the proposed approach.

**Weaknesses:**

Complexity of Presentation: The paper's writing is overly complex. In my view, it essentially presents a post-processing approach where a super-resolution enhancement module is added after compression, and the use of Factorized Fields in the SR module is the main novelty. However, this could have been presented more straightforwardly. The authors attempt to show that this approach benefits both compression and super-resolution, but focusing on one aspect would have been sufficient to convey the key idea. As it stands, the paper lacks a clear focus.

Lack of In-depth Analysis: Although the Factorized Fields concept appears promising, the paper severely lacks comprehensive analysis and in-depth discussion. It fails to clearly illustrate how it differs from and improves upon the original Factor Field. Moreover, there is a lack of explanations regarding why this approach is effective. Substantial in-depth investigations and robust justifications are essential to persuade readers of its superiority and practical value. Without such crucial analysis, it remains arduous to thoroughly understand and truly appreciate the potential of this approach.

**Questions:**

1. What are the values of C_c, H_c and W_c?

2. Further details should be provided to illustrate how Factorized Fields differs from and enhances the original Factor Field.

3. Additionally, there is a lack of explanation regarding the effectiveness of Factorized Fields. Please provide more detailed insights.

---

### Official Review · Reviewer_mF7Y · 2024-11-04

**Soundness:** 3
**Presentation:** 3
**Contribution:** 3
**Rating:** 6
**Confidence:** 2

**Summary:**

The authors propose an approach via factorised fields that utilises a basis-coefficient decomposition to explicitly capture multi-scale visual features and structural components in images, addressing the core challenges of super-resolution and image compression.

**Strengths:**

1. The idea is interesting and offers a comprehensive solution to improve the performance on SR and compression tasks.
2. The quantitative and qualitative results prove that the state-of-the-art performance on the benchmark datasets of the proposed model.

**Weaknesses:**

1.  The paper is confusing sometimes making it difficult to understand. idea of using sawtooth downsampling is not clear. How is it correlating the coordinate transformation and downsampling and can the authors provide its comparisons with other coordinate transformations in the ablation study?
2.  The authors did not discuss limitations of the approach.
3. The authors should provide some more experiments on this basis merging approach for other multi-image restoration  tasks like video SR, burst SR?

**Questions:**

Please check the weaknesses.

---

### Note · Authors · 2024-11-14

I have read and agree with the venue's withdrawal policy on behalf of myself and my co-authors.